

# Characterization of tumor heterogeneity by latent haplotypes: a sequential Monte Carlo approach

Oyetunji E. Ogundijo and Xiaodong Wang

Department of Electrical Engineering, Columbia University, New York, NY, United States of America

## ABSTRACT

Tumor samples obtained from a single cancer patient spatially or temporally often consist of varying cell populations, each harboring distinct mutations that uniquely characterize its genome. Thus, in any given samples of a tumor having more than two haplotypes, defined as a scaffold of single nucleotide variants (SNVs) on the same homologous genome, is evidence of heterogeneity because humans are diploid and we would therefore only observe up to two haplotypes if all cells in a tumor sample were genetically homogeneous. We characterize tumor heterogeneity by latent haplotypes and present state-space formulation of the feature allocation model for estimating the haplotypes and their proportions in the tumor samples. We develop an efficient sequential Monte Carlo (SMC) algorithm that estimates the states and the parameters of our proposed state-space model, which are equivalently the haplotypes and their proportions in the tumor samples. The sequential algorithm produces more accurate estimates of the model parameters when compared with existing methods. Also, because our algorithm processes the variant allele frequency (VAF) of a locus as the observation at a single time-step, VAF from newly sequenced candidate SNVs from next-generation sequencing (NGS) can be analyzed to improve existing estimates without re-analyzing the previous datasets, a feature that existing solutions do not possess.

# INTRODUCTION

Tumors contain multiple, genetically diverse subclonal populations of cells, each subclone harboring distinct mutations that uniquely characterize its genome (*Marusyk & Polyak, 2010*; *Meacham & Morrison, 2013*; *Heppner, 1984*). Tumor subclones often evolve from a single ancestral population (*Hughes et al., 2014*; *Gerlinger et al., 2012*; *Visvader, 2011*; *Nowell, 1976*). The genetic diversities that distinguish these subclones are a direct result of evolutionary processes that drive tumor progression, especially the series of somatic genetic variants which arise stochastically by a sequence of randomly acquired mutations (*Hanahan & Weinberg, 2011*; *Hanahan & Weinberg, 2000*).

Identifying and characterizing tumor subclonality is crucial for understanding the evolution of tumor cells. The knowledge is important for designing more effective treatments for cancer, especially in avoiding cancer relapse and chemotherapy resistance (*Garraway & Lander, 2013*). For instance, research has shown the links

Corresponding author
Xiaodong Wang,
wangx@ee.columbia.edu

between the presence of driver mutations within subclones and the adverse clinical outcomes (*Landau et al., 2013*).

Tumor heterogeneity has been studied using the NGS technology (*Lee et al., 2016*; *Gerlinger et al., 2012*). Somatic mutations are quantified using whole exome sequencing (WES) and whole genome sequencing (WGS) of samples (*Marusyk, Almendro & Polyak, 2012*), and can be explained by differences in genomes of subclones and the varying proportions of these subclones (*Lee et al., 2016*; *Landau et al., 2013*; *Russnes et al., 2011*; *Navin et al., 2010*; *Marusyk & Polyak, 2010*). One method to assess the heterogeneity of a given tumor is to probe individual cell using fluorescent markers (*Navin et al., 2010*; *Irish et al., 2004*) and another is to perform single cell sequencing (*Xu et al., 2012*; *Hou et al., 2012*; *Navin et al., 2011*). These approaches have several limitations that prevent their wider usage in examining and quantifying the level of heterogeneity in a given sample. For instance, evaluating large number of cells by probing them individually can be expensive. Also, the spatial positioning of the cells relative to other cells in the tumor are lost in the process.

In the literature, a few computational methods have been proposed to explain the inherent structure of tumor heterogeneity. For instance, *Larson & Fridley (2013)* and *Su et al. (2012)* viewed a tumor sample as a mixture of tumor cells and normal cells. Although their method can estimate tumor purity levels for paired tumor-normal tissue sample using DNA sequencing data, unpaired and multiple tumor samples are not considered. A more prominent approach is the arrangement of single nucleotide variants (SNVs) in clusters using clustering models such as the Dirichlet Process (DP) (*Roth et al., 2014*; *Jiao et al., 2014*; *Shah et al., 2009*; *Ding et al., 2010*; *Bashashati et al., 2013*). Although the clustered SNVs provide some information about tumor heterogeneity, the inference does not directly identify subclones or haplotypes in the tumor samples.

More recently, *Lee et al. (2016)* and *Xu et al. (2015)* proposed a feature allocation model for estimating tumor heterogeneity by estimating haplotypes and their proportions in the tumor. This model provides insights on how haplotypes may be distributed within a tumor, using WGS data measuring variant allele frequencies (VAFs) at SNVs. Mathematically, the model can be interpreted as blind matrix factorization. A matrix of expected VAFs at SNVs for different samples is decomposed into a binary matrix of haplotypes (with an unknown number of columns, the exact number to be determined by the data), and a matrix of proportions of haplotypes. This model offers certain modeling advantages over the clustering approach: (i) overlapping SNVs can be shared among different subclones, and (ii) non-overlapping SNV clusters (according to the cellular prevalence) are not used as the building block for subclones, i.e., instead of first estimating the SNV clusters and then constructing subclones based on clusters, the model provides a way to infer the subclonal structure based on haplotypes. To make an inference on the haplotypic structure in tumor samples, *Lee et al. (2016)* and *Xu et al. (2015)* proposed a Markov chain Monte Carlo (MCMC-based) and a *maximum a posteriori* (MAP-based) asymptotic derivations (MAD) algorithms respectively. However, if more VAFs are available for newly called SNV(s), both algorithms have to be restarted in order to incorporate the newly called SNV(s). Moreover, MCMC approach in general as previously shown (*Nguyen et al., 2016*; *Jasra, Stephens &*
*Holmes, 2007*) is plagued with some inherent issues which often limit its performance: (i) sometimes, it is difficult to assess when the Markov chain has reached its stationary regime of interest (ii) requirement of burn-in period and thinning interval, and most importantly, (iii) if the target distribution is highly multi-modal, MCMC algorithms can easily become trapped in local modes.

In this paper, we consider the feature allocation modeling approach in analyzing the WGS data measuring VAFs at SNVs, and present an efficient sequential Monte Carlo (SMC) algorithm (*Doucet, De Freitas & Gordon, 2001*; *Doucet, Godsill & Andrieu, 2000*) for estimating the binary matrix of haplotypes and the proportions in the samples. Specifically, we formulate the feature allocation problem using a state-space. We consider the following in our state-space framework: (i) the rows of the haplotype binary matrix are considered as the states of the system, exploiting the sequential construction of a binary matrix with an unknown number of columns using the Indian Buffet Process (IBP), (ii) the proportions matrix and other parameters are considered as the parameters of the model, (iii) the observed VAF at each SNV are processed, for all samples at a time. SMC is a very powerful algorithm that belongs to a broad class of recursive filtering techniques (*Ogundijo, Elmas & Wang, 2017*; *Ogundijo & Wang, 2017*). Instead of processing all the observations at once, observations are processed sequentially, one after the other. The posterior probability density function (PDF) of every state is computed every time a measurement is observed. The posterior distributions of the variables of interest are approximated with a set of properly weighted particles (*Doucet, De Freitas & Gordon, 2001*). With the SMC methods, we can treat, in a principled way, any type of probability distribution, nonlinearity and non-stationarity (*Kitagawa, 1998*; *Kitagawa, 1996*). We compare the proposed SMC algorithm with the existing MCMC-based and MAP-based algorithms. In terms of the accuracy of estimates of the matrix of haplotypes and the matrix of proportions, denoted as **Z** and **W**, our proposed SMC method produces better results.

The remainder of this paper is organized as follows. In 'System Model and Problem Formulation', we describe the system model and problem formulation. We describe the general principle of the SMC filtering algorithms, and then derive our proposed SMC algorithm for estimating the mutational profile of each haplotype and their proportions in the samples, in a sequential fashion. In 'Results and Discussion', we investigate the performance of the proposed method using simulated datasets and the chronic lymphocytic leukemia (CLL) datasets, the real tumor samples obtained from three patients in (*Schuh et al., 2012*). Finally, 'Discussion' concludes the paper.

In this paper, we use the following notations:

1. $p(\cdot)$ and $p(\cdot|\cdot)$ denote a probability density function (PDF) and a conditional PDF, respectively.
2. $P(\cdot)$ and $P(\cdot|\cdot)$ denote a probability and a conditional probability mass function, respectively.
3. $\mathcal{N}(\mu, \sigma^2)$ denotes a Gaussian distribution with mean $\mu$ and variance $\sigma^2$.
4. Binomial$(n, p)$ denotes a binomial distribution with $n$ number of trials and $p$ probability of success. (Binomial$(1, p) = $ Bern$(p)$, i.e, Bernoulli distribution with success probability $p$).

5. Pois($\lambda$) denotes a Poisson distribution with mean parameter $\lambda$.
6. Beta($\alpha_1, \beta_1$) denotes a beta distribution with shape parameters $\alpha_1$ and $\beta_1$.
7. Gam($\alpha_0, \beta_0$) denotes a gamma distribution with shape parameter $\alpha_0$ and rate parameter $\beta_0$.
8. Dir($\boldsymbol{\alpha}$) denotes a Dirichlet distribution with a vector of concentration parameters $\boldsymbol{\alpha}$, and $\hat{x}$ denotes the estimate of variable $x$.

## SYSTEM MODEL AND PROBLEM FORMULATION

In an NGS experiment designed to probe the heterogeneity of a tumor sample, two matrices $\mathbf{Y}$ and $\mathbf{V}$, each with dimension $T \times S$, of count data are often observed, where $y_{ts}$ and $v_{ts}$ denote the elements in the $t^{th}$ row and $s^{th}$ columns of $\mathbf{Y}$ and $\mathbf{V}$, respectively. At the genomic position of SNV $t$ for tissue sample $s$, $y_{ts}$ denotes the number of reads that bear a variant sequence and $v_{ts}$ denotes the total number of reads, $t = 1, ..., T, s = 1, ..., S$. In summary, the datasets are count data for $T$ SNVs and $S$ samples. We follow the binomial sampling framework in (*Lee et al., 2016*; *Zare et al., 2014*; *Xu et al., 2015*) to model the count data:

$$y_{ts} \overset{ind.}{\sim} \text{Binomial}(v_{ts}, p_{ts}), \quad t = 1, ..., T, \ s = 1, ..., S, \tag{1}$$

where $p_{ts}$ are the success probabilities and equivalently the expected VAFs, given by:

$$p_{ts} = w_{0s}p + \sum_{c=1}^{C} z_{tc}w_{cs}, \ t = 1, ..., T, s = 1, ..., S, \tag{2}$$

where $C$ denotes the *unknown number of distinct haplotypes in the tumor samples*, $z_{tc} \in \{0, 1\}$ denotes an indicator of the event that SNV $t$ bears a variant sequence for haplotype $c$ and $w_{cs}$ denotes the proportion of haplotype $c$ in sample $s$. The term $\sum_{c=1}^{C} z_{tc}w_{cs}$ explains $p_{ts}$ as arising from sample $s$ being composed of a mix of hypothetical haplotypes which include a mutation for SNV $t$ ($z_{tc} = 1$), or do not include a mutation for SNV $t$ ($z_{tc} = 0$). In addition, there is a background haplotype $c = 0$, which includes all SNVs. The background haplotype accounts for experimental noise and haplotypes that appear with negligible abundance. The first term in (2) relates to this background haplotype, where $p$ denotes the relative frequency of observing a mutation at an SNV due to noise and artifact, assuming equal frequency for all SNVs, and $w_{0s}$ denotes the proportion in sample $s$ (*Lee et al., 2016*). In (2), if we (i) collect the indicators $z_{tc}$ in an $T \times C$ binary matrix $\mathbf{Z}$, (ii) collect all $p$'s in a $T$-dimensional column vector $\mathbf{p}$ and (iii) collect the proportions $w_{0s}$ and $w_{cs}$ in an $C' \times S$ matrix $\mathbf{W}$ of probabilities, where $C' = C + 1$ and each column of $\mathbf{W}$ sums to unity, then we can write (2) as $\mathbf{P}_{ts} = \mathbf{Z}' \cdot \mathbf{W}$, where $\mathbf{P}_{ts}$ denotes the matrix of success probabilities and equivalently, the matrix of expected VAFs and $\mathbf{Z}' = [\mathbf{p} \ \mathbf{Z}]$. If the expected VAFs are approximated with the observed VAFs, we can directly solve the matrix factorization problem but instead the observed VAFs are modeled with a probability distribution in (1). However, it should be noted that the number of latent haplotypes $C$ is unknown, and this leaves the number of columns in $\mathbf{Z}$ and equivalently, the number of rows in $\mathbf{W}$ unknown, left to be estimated from the data.

Our goal is to perform a joint inference on $C$, $\mathbf{Z}$, $\mathbf{W}$ and $p$, all of which explain the heterogeneity in the tumor samples, using the observed VAFs of SNVs described by the

matrices $\mathbf{Y}$ and $\mathbf{V}$, the input data. To do this, we describe the system using a state-space model and then derive an efficient SMC algorithm to estimate all the hidden states and the model parameters in our model, in a sequential fashion. Our analysis is restricted to mutations in copy-number neutral regions.

## State-space formulation

Our state-space formulation of the problem exploits the sequential construction of $\mathbf{Z}$. Specifically, we consider the $t^{th}$ row of the data matrix $\mathbf{Y}$ and $\mathbf{V}$ as the new observation at *time t* of our state-space model, treat the $t^{th}$ row of the binary matrix $\mathbf{Z}$ as the hidden state at *time t*, and $\mathbf{W}$ and $p$ as the model parameters. Before explicitly stating the state transition and the observation models, we succinctly describe the prior distribution on a "left-ordered" binary matrix (i.e., ordering the columns of the binary matrix from left to right by the magnitude of the binary expressed by that column, taking the first row as the most significant bit) with a finite number of rows and an unknown number of columns. The prior distribution as detailed in (*Griffiths & Ghahramani, 2011*; *Ghahramani & Griffiths, 2006*) is given by:

$$P(\mathbf{Z}) = \frac{\alpha^{C_+}}{\prod_{h=1}^{2^T-1} C_h!} \exp\{-\alpha H_T\} \prod_{c=1}^{C^+} \frac{(T-m_c)!(m_c-1)!}{T!}, \tag{3}$$

where $C_+$ denotes the number of columns of $\mathbf{Z}$ with non-zero entries, $m_c$ denotes the number of 1's in column $c$, $T$ denotes the number of rows in $\mathbf{Z}$, $H_T = \sum_{t=1}^{T} 1/t$ denotes the $T^{th}$ harmonic number, and $C_h$ denotes the number of columns in $\mathbf{Z}$ that when read top-to-bottom form a sequence of 1's and 0's corresponding to the binary representation of the number $h$.

The distribution in (3) can be derived as the outcome of a *sequential generative process* called the *Indian buffet process* (*Griffiths & Ghahramani, 2011*; *Doshi-Velez, 2009*). Imagine that in an Indian buffet restaurant, we have $T$ customers who arrive at the restaurant sequentially, one after the other. The first customer walks into the restaurant and loads her plate from the first $c_1$ dishes, where $c_1 = \text{Pois}(\alpha)$ ($\alpha$ is similar to the dispersion parameter in the Chinese Restaurant Process (*Zhang, 2008*)). The $t^{th}$ customer will choose a particular dish according to the popularity of the dish, i.e., choosing a dish with probability $m_c/t$, where $m_c$ denotes the number of people who have previously chosen the $c^{th}$ dish, and in addition, chooses $\text{Pois}(\alpha/t)$ new dishes as well. Now, if we record the choices of each customer on each row of a matrix, where each column corresponds to a dish on the buffet (1 if the dish is chosen, and 0 if not), then such a binary matrix is a draw from the distribution in (3) (*Ghahramani & Griffiths, 2006*). The entire process is sequential because the choices made by the $t^{th}$ customer are dependent only on the choices made by the $t-1$ preceeding customers and not on the remaining $T-t$ customers.

In our case, the dishes in the IBP are the haplotypes in the tumor samples, the SNVs are the customers and more importantly, the $t^{th}$ customer is the observation at *time t* in our state-space model. Moreover, if we consider $\mathbf{z}_t = [z_{t1}, z_{t2}, ..., z_{tC}]$ in (2), which is equivalently the $t^{th}$ row of $\mathbf{Z}$ as the state at time $t$, then we can write our state transition

model, following the sequential process described by the IBP as follows:

$$P(\mathbf{z}_t|\mathbf{Z}_{t-1},\alpha), \tag{4}$$

where $\mathbf{Z}_{t-1}$ denotes the previous $t-1$ rows in $\mathbf{Z}$. The algorithm to sample from (4) is presented in Algorithm 1 in the Supplemental Information 1. Note that in the algorithm, $\mathbf{Z}_t$ is implicitly constructed from $\mathbf{Z}_{t-1}$ and if in the process, new non-zero column(s) is/are introduced in $\mathbf{Z}_t$ ($\mathrm{Pois}(\alpha/t) > 0$), then new row(s) will be added to $\mathbf{W}$ as well. On the other hand, if the numbers of non-zero columns in $\mathbf{Z}_{t-1}$ and $\mathbf{Z}_t$ are the same, then the number of rows in $\mathbf{W}$ does not change between $t-1$ and $t$. To account for any possible change of dimension in $\mathbf{W}$, we re-parameterize matrix $\mathbf{W}$. Specifically, we rewrite $w_{cs} = \theta_{cs}/\sum_{c'=0}^{C}\theta_{c's}$, which implies that we estimate $\theta_{cs}$ and compute $w_{cs}$ from the estimates of $\theta_{cs}$. This procedure ensures that each column of $\mathbf{W}$ sums to unity at any point in time during the process.

Moreover, since we are interested in the final estimates of the model parameters $\mathbf{W}$ and $p$, we create artificial dynamics for these parameters using the random walk model, i.e.,

$$\begin{aligned}\phi_t &\sim p(\phi_t|\phi_{t-1}) = \mathcal{N}(\phi_{t-1},\sigma^2),\\ \phi_t &\in \{p,\theta_{cs}, c=0,1,...,C, s=1,...,S\},\end{aligned} \tag{5}$$

where $\sigma$ denotes the standard deviation. Hence, (4)–(5) fully describe the system state transition.

Similarly, the observation at time $t$ is given by:

$$\begin{aligned}\mathbf{y}_t \sim P(\mathbf{y}_t|\mathbf{Z}_{1:t},\mathbf{W},p) &= P(\mathbf{y}_t|\mathbf{z}_t,\mathbf{W},p)\\ &= \prod_{s=1}^{S}\mathrm{Binomial}(y_{ts}|v_{ts},p_{ts}),\end{aligned} \tag{6}$$

where $\mathbf{y}_t$ denotes the observation at time $t$ (which is conditionally independent of the previous observations $\mathbf{Y}_{t-1}$ given the state $\mathbf{z}_t$), i.e., the $t^{th}$ row of $\mathbf{Y}$. (6) fully describes the measurement model for the system. Finally, (4)–(6) completely describe our proposed state-space model for estimating the mutational profile and the proportion of each haplotype, and the total number of haplotypes in the tumor samples.

## The SMC algorithm

In this section, we briefly describe the SMC filtering framework that will be employed to estimate the states and the parameters of our state-space model (*Doucet, Godsill & Andrieu, 2000*; *Doucet, De Freitas & Gordon, 2001*). Consider the general dynamic system with hidden state variable $\mathbf{x}_t$, in our case, consisting of discrete variables $\mathbf{z}_t$ and continuous variables $\phi_t$, $\phi_t \in \{p_0^t,\theta_{cs}^t, c=0,1,...,C, s=1,...,S\}$, and measurement variable $\mathbf{y}_t$, where there is an initial state model $p(\mathbf{x}_0)$, and $\forall t \geq 1$, a state transition model given in (4)–(5) and an observation model given in (6). The sequence $\mathbf{X}_t = \{\mathbf{x}_1,\mathbf{x}_2,...,\mathbf{x}_t\}$ is not observed and we want to estimate it for each time $t$, given that the we have the observations $\mathbf{Y}_t = \{\mathbf{y}_1,\mathbf{y}_2,...,\mathbf{y}_t\}$.

Our goal is to approximate the posterior distribution of states $p(\mathbf{X}_t|\mathbf{Y}_t)$ using particles drawn from it. However, getting such particles from $p(\mathbf{X}_t|\mathbf{Y}_t)$ is usually not feasible. We can still implement an estimate using $N$ particles, $\{\mathbf{X}_t^i\}_{i=1}^{N}$, taken from another distribution,

$q(\mathbf{X}_t|\mathbf{Y}_t)$, whose support includes the support of $p(\mathbf{X}_t|\mathbf{Y}_t)$ (importance sampling theorem). For the approximation, the weights associated with the particles are calculated as follows:

$$\tilde{w}_t^i = \frac{p(\mathbf{X}_t|\mathbf{Y}_t)}{q(\mathbf{X}_t|\mathbf{Y}_t)} \quad \text{and} \quad w_t^i = \frac{\tilde{w}_t^i}{\sum_{m=1}^N \tilde{w}_t^m}, \quad i = 1, \dots, N. \tag{7}$$

Thus, the pair $\{\mathbf{X}_t^i, w_{1:t}^i\}_{i=1}^N$ is said to be properly weighted with respect to the distribution $p(\mathbf{X}_t|\mathbf{Y}_t)$, and the approximation $\hat{p}(\mathbf{X}_t|\mathbf{Y}_t)$ is then given by:

$$\hat{p}(\mathbf{X}_t|\mathbf{Y}_t) = \sum_{i=1}^N w_t^i \delta(\mathbf{X}_t - \mathbf{X}_t^i), \quad \text{where } \delta(\mathbf{u}) = \begin{cases} 1, & \text{if } \mathbf{u} = \underline{\mathbf{0}} \\ 0, & \text{otherwise.} \end{cases} \tag{8}$$

Similar to the above importance sampling theory, a sequential algorithm can be obtained as follows. First, we express the full posterior distribution of states $\mathbf{X}_t$ given the observations $\mathbf{Y}_t$ as follows:

$$\begin{aligned} p(\mathbf{X}_t|\mathbf{Y}_t) &\propto p(\mathbf{y}_t|\mathbf{X}_t, \mathbf{Y}_{t-1}) p(\mathbf{X}_t|\mathbf{Y}_{t-1}) \\ &= p(\mathbf{y}_t|\mathbf{X}_t, \mathbf{Y}_{t-1}) p(\mathbf{x}_t|\mathbf{X}_{t-1}, \mathbf{Y}_{t-1}) p(\mathbf{X}_{t-1}|\mathbf{Y}_{t-1}). \end{aligned} \tag{9}$$

At time $t$, we desire to obtain $N$ weighted particles from $p(\mathbf{X}_t|\mathbf{Y}_t)$, which is not feasible. Instead, we define an importance distribution $q(\mathbf{X}_t|\mathbf{Y}_t) = q(\mathbf{x}_t|\mathbf{X}_{t-1}, \mathbf{Y}_t) q(\mathbf{X}_{t-1}|\mathbf{Y}_{t-1})$, where particles can be obtained from, and then calculate the associated unnormalized importance weights as follows:

$$\tilde{w}_t^i = \frac{p(\mathbf{y}_t|\mathbf{X}_t^i, \mathbf{Y}_{t-1}) p(\mathbf{x}_t^i|\mathbf{X}_{t-1}^i, \mathbf{Y}_{t-1})}{q(\mathbf{x}_t^i|\mathbf{X}_t^i, \mathbf{Y}_t)} \frac{p(\mathbf{X}_{t-1}^i|\mathbf{Y}_{t-1})}{q(\mathbf{X}_{t-1}^i|\mathbf{Y}_{t-1})}. \tag{10}$$

Assuming that at time $t-1$, we have already drawn the particles $\{\mathbf{X}_{t-1}^i\}_{i=1}^N$ from the importance distribution $q(\mathbf{X}_{t-1}|\mathbf{Y}_{t-1})$ and the corresponding normalized weights written as follows:

$$w_{t-1}^i \propto \frac{p(\mathbf{X}_{t-1}^i|\mathbf{Y}_{t-1})}{q(\mathbf{X}_{t-1}^i|\mathbf{Y}_{t-1})}, \quad i = 1, \dots, N, \tag{11}$$

we can now draw particles $\{\mathbf{X}_t^i\}_{i=1}^N$ from the importance distribution $q(\mathbf{X}_t|\mathbf{Y}_t)$ by drawing the new state particles for the time step $t$ as $\mathbf{x}_t^i \sim q(\mathbf{x}_t|\mathbf{X}_{t-1}^i, \mathbf{Y}_t)$, and write $\{\mathbf{X}_t^i\}_{i=1}^N = \{\mathbf{x}_t^i, \mathbf{X}_{t-1}^i\}_{i=1}^N$. If we substitute (11) into (10), the weights at time $t$ satisfy the recursion:

$$\tilde{w}_t^i \propto w_{t-1}^i \frac{p(\mathbf{y}_t|\mathbf{X}_t^i, \mathbf{Y}_{t-1}) p(\mathbf{x}_t^i|\mathbf{X}_{t-1}^i, \mathbf{Y}_{t-1})}{q(\mathbf{x}_t^i|\mathbf{X}_t^i, \mathbf{Y}_t)}, \quad i = 1, \dots, N, \tag{12}$$

and then the weights are normalized to sum to unity.

So far, we have presented a generic sequential sampling algorithm. We obtain the optimal importance distribution by setting $q(\mathbf{x}_t^i|\mathbf{X}_{t-1}^i, \mathbf{Y}_t) = p(\mathbf{x}_t^i|\mathbf{X}_{t-1}^i, \mathbf{Y}_t)$, and the weights in (12) become $\tilde{w}_t^i \propto w_{t-1}^i p(\mathbf{y}_t|\mathbf{X}_{t-1}^i, \mathbf{Y}_{t-1})$ (*Ristic, Arulampalam & Gordon, 2004*) i.e., if the distributions $p(\mathbf{y}_t|\mathbf{X}_{t-1}^i, \mathbf{Y}_{t-1})$ and $p(\mathbf{x}_t^i|\mathbf{X}_{t-1}^i, \mathbf{Y}_{t-1})$ are conjugates, then closed form solutions can be obtained for $p(\mathbf{x}_t^i|\mathbf{X}_{t-1}^i, \mathbf{Y}_t)$, and hence, $p(\mathbf{y}_t|\mathbf{X}_{t-1}^i, \mathbf{Y}_{t-1})$. However, if no such conjugacy exists, which is the case for our state-space model, the most popular choice and

---

**Algorithm 1** SMC Algorithm for Characterizing Tumor Heterogeneity

---

**Input: Y, V.**

1: Initialize $N$ particles $\{\mathbf{z}_0^i, p_0^i, \mathbf{W}_0^i\}_{i=1}^N$

2: **for** $t = 1, \ldots, T$ **do**

3:      **for** $i = 1, \ldots, N$ **do**

4:          Sample $\mathbf{z}_t^i$ from $\mathbf{Z}_{t-1}^i$ using **Algorithm 1** in the Supplementary Material.

5:          $n_1 \leftarrow$ number of columns in $\mathbf{Z}_{t-1}^i$

6:          $n_2 \leftarrow$ length of $\mathbf{z}_t^i$

7:          $d \leftarrow (n_2 - n_1)$

8:          **if** $d = 0$ **then**

9:

$$\mathbf{Z}_t^i \leftarrow \begin{bmatrix} \mathbf{Z}_{t-1}^i \\ \mathbf{z}_t^i \end{bmatrix}$$

10:             Sample $\mathbf{W}_t^i$ using (5)

11:          **else**

12:

$$\mathbf{Z}_t^i \leftarrow \begin{bmatrix} \mathbf{Z}_{t-1}^i & \mathbf{0} \\ \mathbf{z}_t^i \end{bmatrix}$$

13:             Sample $\mathbf{W}_t^i$ using (5)

14:             Sample new rows of $\mathbf{W}_t^i$ from the priors in (14)

15:          **end if**

16:          Calculate $\tilde{w}_t^i$ using (13)

17:      **end for**

18:      Normalize the weights

19:      Perform resampling

20: **end for**

21: Approximations of posterior estimates of all the unknown variables are obtained from the final particles and weights, using the procedures highlighted in (Lee et al., 2016) and discussed in the Supplementary Material.

---

equally efficient solution (*Van Der Merwe, 2004*) is to set $q(\mathbf{x}_t^i | \mathbf{X}_{t-1}^i, \mathbf{Y}_t) = p(\mathbf{x}_t^i | \mathbf{X}_{t-1}^i)$ (in (4)–(5)) (*Wood & Griffiths, 2007*; *Särkkä, 2013*). Considering the assumed independence in our model, i.e., $p(\mathbf{x}_t^i | \mathbf{X}_{t-1}^i, \mathbf{Y}_{t-1}) = p(\mathbf{x}_t^i | \mathbf{X}_{t-1}^i)$ and $p(\mathbf{y}_t | \mathbf{X}_t^i, \mathbf{Y}_{t-1}) = p(\mathbf{y}_t | \mathbf{x}_t^i)$, then (12) becomes:

$$\tilde{w}_t^i \propto w_{t-1}^i p(\mathbf{y}_t | \mathbf{x}_t^i) = w_{t-1}^i p(\mathbf{y}_t | \mathbf{z}_t^i, \mathbf{W}_t^i), \tag{13}$$

and the weights are normalized. Such implementation is commonly referred to as a bootstrap filter in the literature (*Särkkä, 2013*).

However, the variance of the weights increases over time, a condition referred to as degeneracy in the literature (*Doucet, De Freitas & Gordon, 2001*). To avoid this, we perform resampling, at every time step, owing to the choice of the importance distribution (*Wood*
_______________

*& Griffiths, 2007*; *Särkkä, 2013*), discarding the ineffective particles and multiplying the effective ones. The resampling procedure (*Särkkä, 2013*) is described in the Supplemental Information 1.

Finally, our proposed SMC algorithm for estimating the mutational profiles and the proportions of the haplotypes in the tumor samples i.e., the states and the parameters of our state-space model, is presented in Algorithm 1. The algorithm is initialized by taking particles from the prior distributions of the parameters. We assume the following:

$$
\begin{aligned}
\theta_{0s} &\overset{i.i.d}{\sim} \text{Gamma}(a_0, 1), \ s = 1, ..., S, \ p \sim \text{Beta}(a_{00}, b_{00}) \\
\theta_{cs} &\overset{i.i.d}{\sim} \text{Gamma}(a_1, 1), \ s = 1, ..., S, c = 1, ..., C,
\end{aligned}
\tag{14}
$$

such that $w_{cs} = \theta_{cs} / \sum_{c'=0}^{C} \theta_{c's}$ and consequently, $\sum_{c'=0}^{C} w_{c's} = 1$ and assume that $a_{00} << b_{00}$ to impose a small $p$. We report the posterior estimates of all the unknown variables using the procedure highlighted in *Lee et al. (2016)*, with the details discussed in the Supplemental Information 1.

## RESULTS AND DISCUSSION

In this section, we demonstrate the performance of the proposed SMC algorithm using both simulated datasets and the CLL datasets obtained from three different patients (*Schuh et al., 2012*). In addition, we compare the estimates obtained from the proposed SMC algorithm with the MCMC-based algorithm proposed in *Lee et al. (2016)* and the MAP-based algorithm proposed in *Xu et al. (2015)*. For the MCMC-based algorithm, the algorithm parameters are set as in *Lee et al. (2016)*, running a simulation over $40,000$ iterations, discarding the first $15,000$ iterations as burn-in. For the MAD-based algorithm, we ran $1,000$ random initializations for each dataset.

### Simulated data

We produced simulated datasets with average sequencing depth $r \in \{20, 40, 50, 200, 100, 10,000\}$ per locus. We fixed the number of haplotypes $C = 4$ and number of samples $S \in \{1, 3, 5, 10, 20\}$. For all combinations of $r$ and $S$, we generated the variants count matrix $\mathbf{Y}$ and the total count matrix $\mathbf{V}$ for different number of SNVs $T \in \{20, 60\}$. Specifically, we generated each entry of $\mathbf{V}$, i.e., $v_{ts}$ from $\text{Pois}(r)$ and to generate each entry of $\mathbf{Y}$, i.e., $y_{ts}$, we did the following: (i) generate each column of $\mathbf{W}$ from $\text{Dir}([a_0, a_1, ..., a_4])$, where $a_0 = 0.2$, and $a_c$, $c \in \{1, ..., 4\}$ is randomly chosen from the set $\{2, 4, 5, 6, 7, 8\}$, (ii) generate entries of $\mathbf{Z}$ independently from $\text{Bern}(0.6)$, (iii) set $p = 0.02$, (iv) compute $p_{ts}$ using (2), and finally, (v) generate $y_{ts}$ as an independent sample from $\text{Binomial}(v_{ts}, p_{ts})$.

Next, we run the proposed SMC-based, MCMC-based and the MAP-based algorithms on the simulated $\mathbf{Y}$ and $\mathbf{V}$ datasets. To quantify the performance of the algorithms, we define the following metrics: haplotype error ($e_Z$), proportion error ($e_W$) and the error of the success probabilities ($e_{p_{ts}}$) as follows:

$$
e_Z = \frac{1}{TC} \sum_{t=1}^{T} \sum_{c=1}^{C} |\hat{z}_{tc} - z_{tc}|, \ e_W = \frac{1}{CS} \sum_{c=0}^{C} \sum_{s=1}^{S} |\hat{w}_{cs} - w_{cs}|,
$$

**Table 1** $e_{p_{ts}}$, $e_Z$ and $e_W$ computed for the proposed SMC-based, MCMC-based and the MAP-based algorithms for $T = 60$, $C = 4$, $S = 10$ and $r \in \{20, 40, 50, 200, 1{,}000, 10{,}000\}$.

| | $T = 60$, $C = 4$ and $S = 10$ | | | | | | | | |
|---|---|---|---|---|---|---|---|---|---|
| | SMC-based | | | MCMC-based | | | MAP-based | | |
| $r$ | $e_{p_{ts}}$ | $e_Z$ | $e_W$ | $e_{p_{ts}}$ | $e_Z$ | $e_W$ | $e_{p_{ts}}$ | $e_Z$ | $e_W$ |
| 20 | 0.0200 | 0.1033 | 0.0416 | 0.1240 | 0.1200 | 0.1001 | 0.1021 | 0.1400 | 0.0900 |
| 40 | 0.0137 | 0.0033 | 0.0316 | 0.0638 | 0.0536 | 0.0422 | 0.0490 | 0.0505 | 0.0388 |
| 50 | 0.0148 | 0.0017 | 0.0173 | 0.0745 | 0.0404 | 0.0601 | 0.0820 | 0.0500 | 0.0428 |
| 200 | 0.0122 | 0.0000 | 0.0107 | 0.0219 | 0.0325 | 0.0200 | 0.0301 | 0.0305 | 0.0211 |
| 1000 | 0.0100 | 0.0000 | 0.0199 | 0.0302 | 0.0100 | 0.0324 | 0.0411 | 0.0250 | 0.0220 |
| 10,000 | 0.0012 | 0.0000 | 0.0020 | 0.0100 | 0.0050 | 0.0100 | 0.0110 | 0.0105 | 0.0111 |

and

$$e_{p_{ts}} = \frac{1}{TS} \sum_{t=1}^{T} \sum_{s=1}^{S} |\hat{p}_{ts} - p_{ts}|, \text{ where } \hat{p}_{ts} = \hat{p}\hat{w}_{0s} + \sum_{c=1}^{C} \hat{z}_{tc} \hat{w}_{cs}.$$

However, since this is a blind decomposition, one does not know a priori which column of $\hat{Z}$ corresponds to which column of $Z$. To resolve this, we calculate $e_Z$ with every permutation of the columns of $\hat{Z}$ and then select the permutation that results in the smallest $e_Z$. The selected permutation is then used in computing $e_W$ and $e_{p_{ts}}$.

The results obtained from the analyses of the simulated datasets are presented in Table 1, Figs. 1–3 and Table 1 in the Supplemental Information 1. Table 1 shows $e_{p_{ts}}$, $e_Z$ and $e_W$ obtained for the datasets from $T = 60$ SNVs, $C = 4$ haplotypes and $S = 10$ samples for all the average sequencing depth $r \in \{20, 40, 50, 200, 1{,}000, 10{,}000\}$. Similar results are presented in the Supplemental Information 1 with $T = 20$ SNVs and $S = 5$ samples. From the results obtained for the different number of average sequencing depth $r$ and number of SNVs, the proposed SMC algorithm yields more accurate estimates of the model parameters when compared with the other two algorithms. Specifically, the SMC-based algorithm produced lower error values $e_{p_{ts}}$, $e_Z$ and $e_W$ in all the datasets analyzed. Moreover, we investigated the effect of sample size on the proposed SMC algorithm. As shown in Figs. 1–3, the results show slight improvement as the the number of samples increased. It can be observed that the results are less sensitive to sample size when $S > 5$. Also noticed is a slight improvement in the results when the average sequencing depth $r$ is increased.

By construction, the proposed SMC algorithm can handle datasets with any number of loci since the VAF of each loci is processed as an observation at every time step. Apart from the ability to process any number of loci, this property allows the proposed algorithm to process VAFs data from newly sequenced candidate SNVs to improve existing estimates without re-analyzing the previous datasets. To validate this, we generated datasets for 1,000 and 2,000 loci respectively and these datasets are analyzed with the proposed SMC algorithm with the results presented in Table 2. In fact, the proposed SMC algorithm

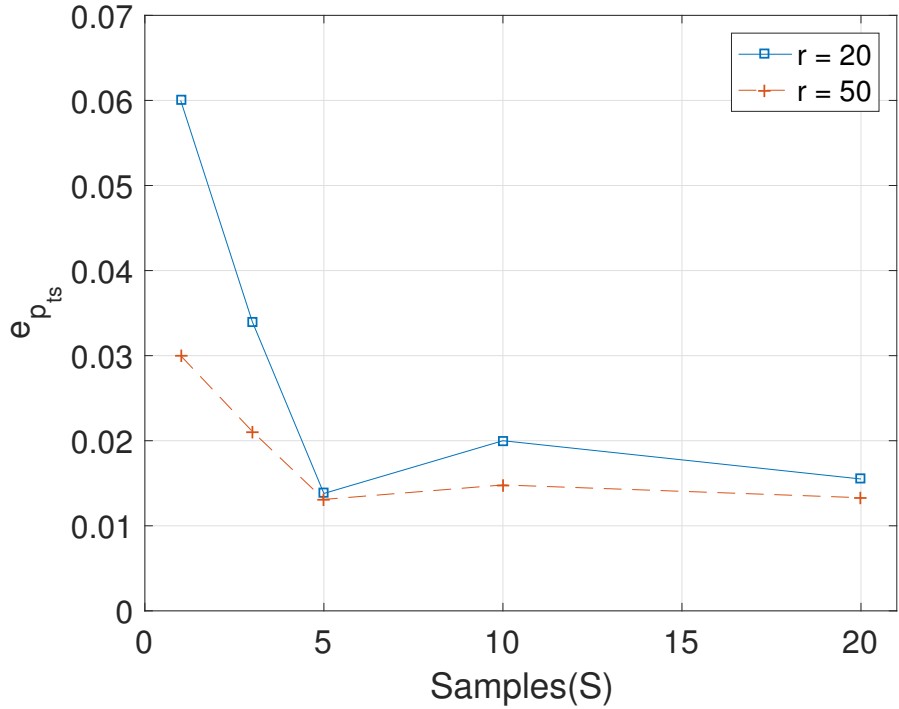

**Figure 1** Plot of $e_{p_{ts}}$ versus sample size $S$ for SNVs $T = 60$, sequencing depth averages $r$ in {20, 50}, and haplotypes $C = 4$.

**Table 2** $e_Z$, $e_W$ and $e_{p_{ts}}$ computed for the proposed SMC algorithm for $C = 4$, $S = 3$ and $T \in \{1,000, 2,000\}$.

| Number of loci ($T$) | Number of samples ($S$) | $e_Z$ | $e_W$ | $e_{p_{ts}}$ |
|---|---|---|---|---|
| 1,000 | 3 | 0.0000 | 0.0060 | 0.0073 |
| 2,000 | 3 | 0.0080 | 0.0048 | 0.0057 |

benefits from larger number of loci because the larger the number of loci, the better the estimate of the proportions. This result is evident from the proportion errors in Table 2.

Lastly, we record the runtime ($t_r$) for the two algorithms on a 3.5 Ghz Intel 8 processors running MATLAB when analyzing some of the datasets described in Table 1 (i.e., $T = 60$, $C = 4$, $S = 10$ and $r = 20$). Observed $t_r$ was 311 seconds and 585 seconds for the proposed SMC and the MCMC-based algorithms, respectively. For the MAP-based algorithm, a single run is 5 seconds but for a set of input data, the algorithm requires different random initializations.

## Real tumor samples: CLL datasets

We evaluate the proposed SMC algorithm on the datasets for the B-cell chronic lymphocytic leukemia (CLL), obtained from three patients: **CLL003**, **CLL006**, and **CLL077** (*Schuh et al., 2012*). These datasets represent the molecular changes in pre-treatment, post-treatment, and relapse samples in the three selected patients, i.e., the samples were taken *temporally*

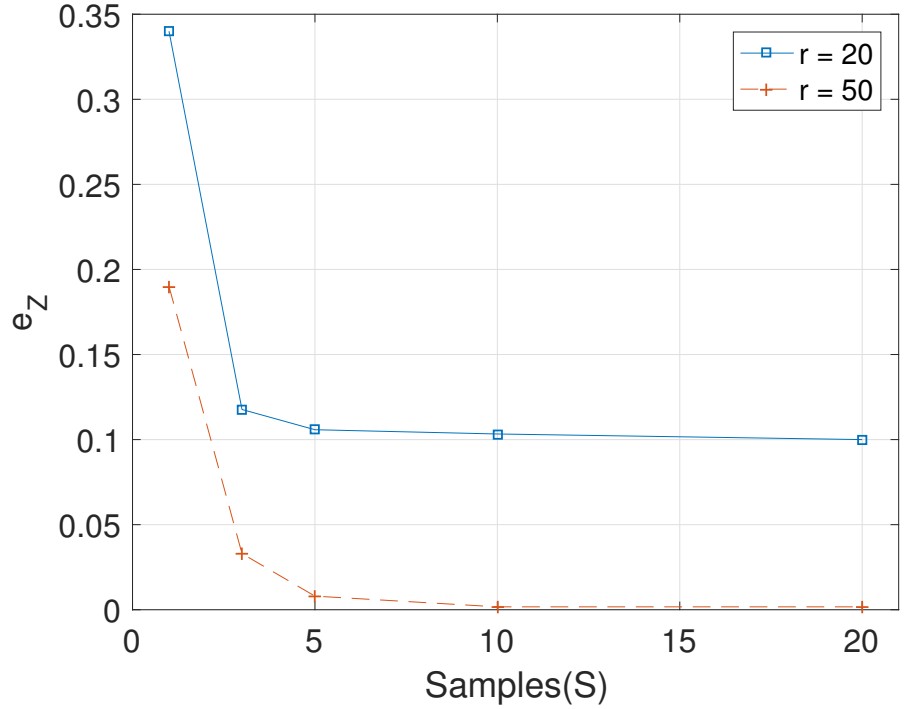

**Figure 2** Plot of $e_Z$ versus sample size $S$ for SNVs $T = 60$, sequencing depth averages $r$ in {20, 50}, and haplotypes $C = 4$.

(see the Supplemental Information 1 for the summary of data pre-processing). The datasets are analyzed with the proposed SMC algorithm and the two other algorithms.

### CLL003

The CLL dataset obtained from patient **CLL003** has 20 distinct loci, shown in the first column in Table 3, and the dataset is analyzed with the proposed SMC algorithm. In Table 3, we present the posterior point estimate of the mutational profiles of the haplotypes in each of the 5 samples, where 1 and 0 denote the variant and the reference sequence, respectively. Moreover, in Fig. 4, we present a graphical representation of how the haplotypes are distributed across the samples. For instance, haplotype $C2$ with approximately 40 percent abundance in the first sample has reduced to approximately 3 percent after the last treatment. In the Supplemental Information 1, we present the table of proportions. The first row on the table and equivalently $C0$ in Fig. 4 comprises of the proportion of the background haplotype, which accounts for experimental noise in each sample. From Table 3, we found that each sample consists of at least 2 dominant haplotypes. For instance, tumor sample **a** is dominated by haplotypes $C2$ and $C6$, each with a proportion of approximately 0.4. Also, we analyzed the same dataset with the other two algorithms and the results are in the Supplemental Information 1.

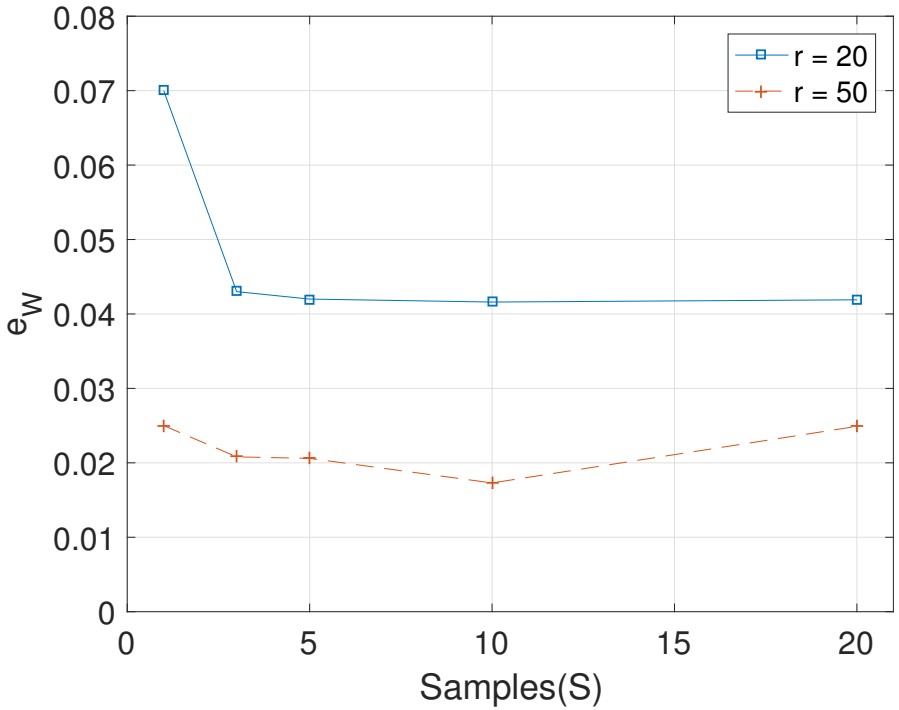

**Figure 3** Plot of $e_W$ versus sample size $S$ for SNVs $T = 60$, sequencing depth averages $r \in \{20, 50\}$, and haplotypes $C = 4$.

### CLL077

The CLL dataset obtained from patient **CLL077** has 16 distinct loci, shown in the first column in Table 4, and the dataset is analyzed with the proposed SMC algorithm. In Table 4, we present the posterior point estimate of mutational profiles of the haplotypes in each of the 5 samples. Moreover, in Fig. 5, we present a graphical representation of how the haplotypes are distributed across the samples, with a table of proportions presented in the Supplemental Information 1. From our analysis results, we found that each sample consists of at least two dominant haplotypes. Also, we analyzed the same dataset with the other two algorithms and the results are in the Supplemental Information 1.

### CLL006

Here, we analyze the CLL dataset obtained from patient **CLL006**. The dataset comprises of 11 loci as shown in Table 5 in the first column, and is analyzed with the proposed SMC algorithm. Table 5 and Fig. 6 show the estimates of mutational profiles and proportions of the haplotypes, respectively. Also, in the Supplemental Information 1, we present the results obtained from analyzing the dataset with the two other algorithms.

As presented in the Supplemental Information 1, the results obtained from the other two algorithms for all the patients are similar. When the estimated haplotypes are compared with some methods that estimate the mutational profiles of tumor subclones, Phylosub proposed in (*Jiao et al., 2014*) and the manual method proposed in (*Schuh et al., 2012*), we found that some of the haplotypes, specifically, $C1, C2, C3$ in **CLL003**; $C3, C4, C5$
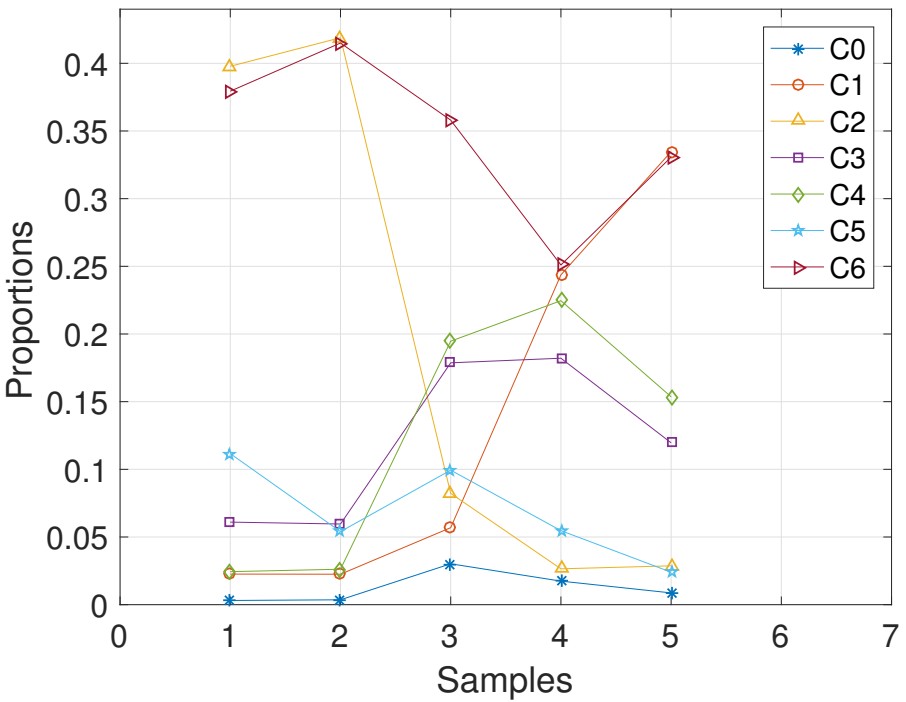

**Figure 4  CLL003: plot of the estimates of the proportions of the haplotypes in each sample.** Samples a, b, c, d, e are designated as 1, 2, 3, 4, 5, respectively.

in **CLL077** and $C1, C2, C3$ in **CLL006**, carry the same set of unique mutations that are present in distinct genomes of subclones. The complete results of the clonal analyses of these methods are presented in the Supplemental Information 1.

Finally, the results presented so far indicate that each heterogeneous tumor sample is made up of more than two haplotypes: usually a few dominant haplotypes and other minor types. The multiple number of haplotypes in a tumor is an indication of the presence of heterogeneity in the sample.

## DISCUSSION

Tumor samples that are obtained from cancer patients often comprise of genetically diverse populations of cells and this defines the heterogeneous nature of the samples. Most of the time, to explain the inherent heterogeneity in the tumor tissues, biologists obtain VAFs datasets via the NGS technology and fit the data into an appropriate model. In this paper, to analyze the observed VAFs data, we employed the feature allocation model. The model, which describes the distribution of haplotypes within the tumor samples, posits that because humans are diploids, having more than two haplotypes in the tumor sample is an evidence of heterogeneity within the sample. According to this model, haplotypes in the tumor samples are the features and SNVs are the experimental units that select the features. So given the observed VAFs of the SNVs, estimating the unknown latent features and the proportions in the samples completely described the inherent heterogeneity in the data.

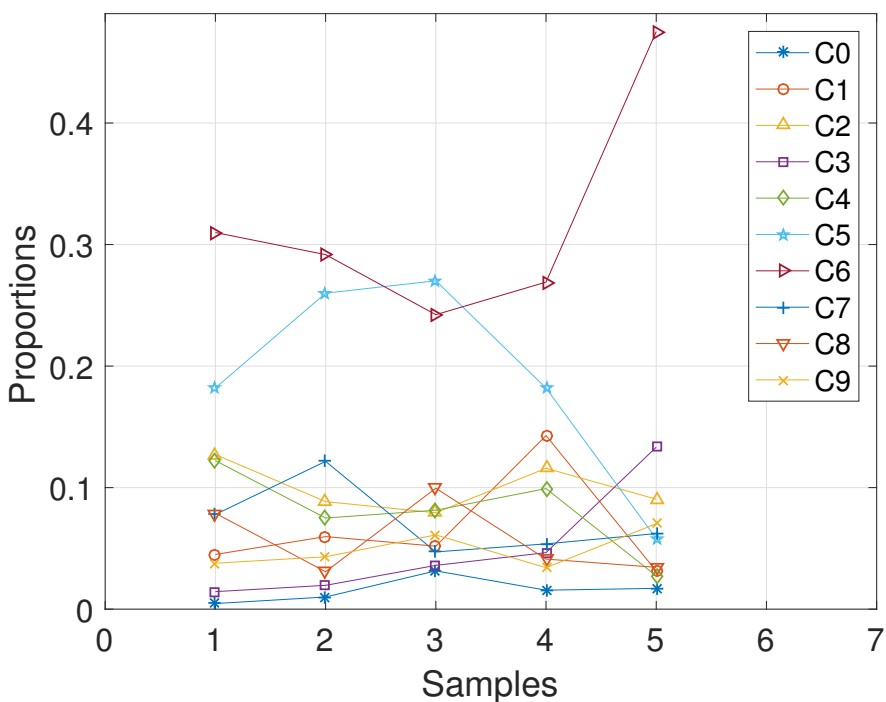

**Figure 5   CLL077: plot of the estimates of the proportions of the haplotypes in each sample.** Samples a, b, c, d, e are designated as 1, 2, 3, 4, 5, respectively.

To estimate the unknown variables in the model, we reformulate the latent feature model into state-space model and present an efficient SMC algorithm. The SMC algorithm takes advantage of the sequential construction of the latent binary matrix, with an unknown number of columns, using the IBP, and treating other variables in the latent feature model as the parameters of our newly formed state-space model. This way, we are able to analyze the VAFs of a single SNV at each iteration. We evaluated our proposed SMC algorithm on simulated datasets, specifically, by varying the average sequencing depth ($r$), the number of tumor samples ($S$) and the number of SNVs ($T$). Also, we analyzed real datasets, i.e., the CLL datasets obtained from (*Schuh et al., 2012*) for three patients. The proposed SMC algorithm produced satisfying results on all categories of datasets analyzed.

Further, we compared the estimates obtained from the proposed SMC-based, MCMC-based and MAP-based algorithms. In terms of the accuracy of estimates, the proposed SMC algorithm yields an improved performance over the two competing algorithms. In the proposed SMC-based algorithm, it is possible to easily incorporate datasets from newly sequenced SNVs (when available) so as to refine the existing estimates. However, in the competing algorithms, to incorporate the new datasets, the entire datasets (old and new) need be analyzed.

In our experiments, we set $N = 500$ particles for all the simulated datasets and for the tumor datasets, we set $N = 1,000$. Also, we run the SMC algorithm five times for the simulated data and 10 times for the CLL datasets. Multiple runs allow the VAFs of each

**Table 3** *CLL003*: estimates of the mutational profiles of haplotypes, Z in the samples.

| Gene | C1 | C2 | C3 | C4 | C5 | C6 |
|---|---|---|---|---|---|---|
| ADAD1 | 1 | 1 | 1 | 0 | 0 | 0 |
| AMTN | 0 | 1 | 0 | 0 | 0 | 0 |
| APBB2 | 0 | 1 | 0 | 0 | 0 | 0 |
| ASXL1 | 1 | 0 | 0 | 1 | 0 | 0 |
| ATM | 0 | 1 | 0 | 0 | 1 | 0 |
| BPIL2 | 0 | 1 | 0 | 0 | 0 | 0 |
| CHRNB2 | 1 | 0 | 0 | 1 | 0 | 0 |
| CHTF8 | 1 | 1 | 1 | 0 | 0 | 0 |
| FAT3 | 1 | 0 | 0 | 1 | 0 | 0 |
| HERC2 | 1 | 1 | 1 | 0 | 0 | 0 |
| IL11RA | 1 | 1 | 1 | 0 | 0 | 0 |
| MTUS1 | 0 | 1 | 0 | 0 | 0 | 0 |
| MUSK | 1 | 0 | 0 | 1 | 0 | 0 |
| NPY | 1 | 0 | 0 | 1 | 0 | 0 |
| NRG3 | 1 | 0 | 0 | 1 | 0 | 0 |
| PLEKHG5 | 0 | 1 | 0 | 0 | 0 | 0 |
| SEMA3E | 1 | 0 | 0 | 1 | 0 | 0 |
| SF3B1 | 1 | 1 | 1 | 0 | 0 | 0 |
| SHROOM1 | 1 | 1 | 1 | 0 | 0 | 0 |
| SPTAN1 | 0 | 1 | 0 | 0 | 0 | 0 |

**Notes.**
The genes where the mutations are found are shown in the first column.

**Table 4** *CLL077*: estimates of the mutational profiles of haplotypes, Z in the samples.

| Gene | C1 | C2 | C3 | C4 | C5 | C6 | C7 | C8 | C9 |
|---|---|---|---|---|---|---|---|---|---|
| BCL2L13 | 1 | 0 | 1 | 1 | 1 | 0 | 1 | 0 | 0 |
| COL24A1 | 0 | 0 | 1 | 0 | 0 | 0 | 0 | 0 | 0 |
| DAZAP1 | 0 | 0 | 0 | 1 | 1 | 0 | 0 | 1 | 0 |
| EXOC6B | 0 | 0 | 0 | 1 | 1 | 0 | 0 | 0 | 1 |
| GHDC | 0 | 0 | 0 | 1 | 1 | 0 | 0 | 0 | 1 |
| GPR158 | 1 | 0 | 1 | 1 | 1 | 0 | 0 | 0 | 1 |
| HMCN1 | 0 | 0 | 1 | 0 | 0 | 0 | 0 | 0 | 0 |
| KLHDC2 | 0 | 0 | 1 | 0 | 0 | 0 | 0 | 0 | 0 |
| LRRC16A | 0 | 0 | 0 | 0 | 1 | 0 | 0 | 0 | 0 |
| MAP2K1 | 0 | 0 | 1 | 0 | 0 | 0 | 0 | 0 | 0 |
| NAMPT | 1 | 0 | 1 | 1 | 1 | 0 | 1 | 0 | 0 |
| NOD1 | 0 | 0 | 1 | 0 | 0 | 0 | 0 | 0 | 0 |
| OCA2 | 0 | 0 | 0 | 1 | 1 | 0 | 0 | 0 | 1 |
| PLA2G16 | 0 | 0 | 0 | 1 | 1 | 0 | 1 | 0 | 0 |
| SAMHD1 | 0 | 1 | 1 | 1 | 1 | 0 | 1 | 0 | 0 |
| SLC12A1 | 0 | 1 | 1 | 1 | 1 | 0 | 0 | 0 | 0 |
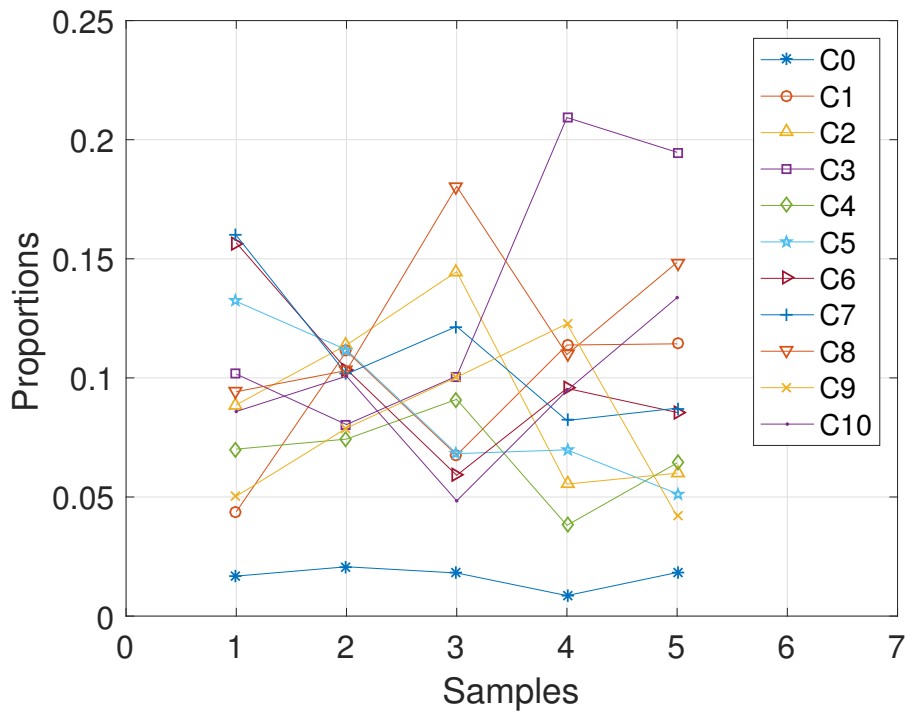

**Figure 6 CLL006: plot of the estimates of the proportions of the haplotypes in each sample.** Samples a, b, c, d, e are designated as 1, 2, 3, 4, 5, respectively.

**Table 5 *CLL006*: estimates of the mutational profiles of haplotypes, Z in the samples.**

| Gene | C1 | C2 | C3 | C4 | C5 | C6 | C7 | C8 | C9 | C10 |
|---|---|---|---|---|---|---|---|---|---|---|
| ARHGAP29 | 1 | 1 | 1 | 1 | 0 | 1 | 0 | 0 | 0 | 0 |
| EGFR | 1 | 1 | 1 | 1 | 1 | 0 | 0 | 0 | 0 | 0 |
| IRF4 | 1 | 0 | 1 | 0 | 0 | 0 | 0 | 0 | 0 | 0 |
| KIAA0182 | 1 | 1 | 1 | 0 | 1 | 0 | 1 | 0 | 0 | 0 |
| KIAA0319L | 1 | 0 | 1 | 1 | 0 | 0 | 0 | 1 | 0 | 0 |
| KLHL4 | 1 | 1 | 1 | 0 | 1 | 1 | 1 | 1 | 0 | 1 |
| MED12 | 1 | 1 | 1 | 1 | 1 | 1 | 1 | 0 | 1 | 0 |
| PILRB | 1 | 1 | 1 | 0 | 1 | 0 | 1 | 0 | 0 | 0 |
| RBPJ | 1 | 0 | 0 | 0 | 0 | 0 | 0 | 0 | 0 | 0 |
| SIK1 | 1 | 1 | 1 | 1 | 1 | 0 | 0 | 0 | 0 | 0 |
| U2AF1 | 1 | 1 | 1 | 0 | 1 | 0 | 0 | 0 | 0 | 0 |

**Notes.**
The genes where the mutations are found are shown in the first column.

SNV to be visited more than once, and we noticed that this, in a way, improves the results of the SMC algorithm.

Finally, we have demonstrated the efficacy of the SMC algorithm, an algorithm that can effectively handle any type of probability distribution, nonlinearity and non-stationarity, particularly in analyzing VAFs of SNVs from tumor samples.

### Funding

This work was supported by the Petroleum Technology Development Fund, Nigeria. The funders had no role in study design, data collection and analysis, decision to publish, or preparation of the manuscript.

### Competing Interests

The authors declare there are no competing interests.

### Author Contributions

- Oyetunji E. Ogundijo conceived and designed the experiments, performed the experiments, analyzed the data, contributed reagents/materials/analysis tools, prepared figures and/or tables, authored or reviewed drafts of the paper, approved the final draft.
- Xiaodong Wang conceived and designed the experiments, contributed reagents/materials/analysis tools, authored or reviewed drafts of the paper, approved the final draft.

### Data Availability

Schuh A, Becq J, Humphray S, Alexa A, Burns A, Clifford R, Feller SM, Grocock R, Henderson S, Khrebtukova I, Kingsbury Z, Luo S, McBride D, Murray L, Menju T, Timbs A, Ross M, Taylor J, Bentley D. 2012. Monitoring chronic lymphocytic leukemia progression by whole genome sequencing reveals heterogeneous clonal evolution patterns. Blood 120:4191-4196; doi: https://doi.org/10.1182/blood-2012-05-433540;

GitHub: https://github.com/moyanre/tumor_haplotypes

### Supplemental Information

Supplemental information for this article can be found online at http://dx.doi.org/10.7717/peerj.4838#supplemental-information.

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
