# Peer review of "Characterization of tumor heterogeneity by latent haplotypes: a sequential Monte Carlo approach"

_PeerJ, doi:10.7717/peerj.4838_

## Round 0.1 · original submission · Major Revisions

Please consider and address the various reviewer comments. In particular, improvements to the writing/language should be incorporated. Comparison to other methods is also a important aspect.

Reviewer 1 ·

Basic reporting

The English could be improved - some sentences are difficult to follow.

Some examples include:

In the abstract, the authors say “..and the parameters of our proposed state-state model.” I assume they mean “state-space model.”

Lines 52-54: Use of both “Although” and “however” makes this sentence confusing.
Line 82: authors should define “IBP” (Indian Buffet Process) in the main text.
Line 93: authors introduce matrices Z and W with no context or definition
Line 112: “SOME LATEX EXAMPLES”; I’m not sure why this is here.

Also, citation of Wersto et al. does not make sense (Line 41). The context involves studies that investigated tumor heterogeneity using NGS technology – Wersto et al. looked at tumor DNA content using flow cytometry. Additionally, the necessary depth of sequencing necessary to characterize somatic variation is quite high, making it rather dubious to claim such studies have been conducted for the “past few decades”.

Experimental design

There are other similar approaches that were not addressed. The most obvious comparator would be the MAD-Bayes approach proposed by Xu et al. (JASA 2015), which similarly touts computational benefits over MCMC-based inference and applies IBP priors for feature allocation. It would seem important to compare against this approach. However, it is not mentioned in the manuscript and its unclear why the authors did not investigate.

Validity of the findings

The methods appear well-outlined and the modes of evaluating error in the simulations seem reasonable.

Reviewer 2 ·

Basic reporting

Some typos must be fixed. Some sentences are too long.

Experimental design

Will the code to reproduce your results be publically available? How can other research teams apply your algorithm?

Validity of the findings

The findings on the synthetic data show some improvement over MCMC, but there are many alternative approaches to compare with. Also, no validation on the real data.

Additional comments

=== Summary:
The manuscript describes an approach to estimate the clonal structure of a tumor using multiple samples obtained from a single cancer that vary patient spatially or temporally. This work may be have some methodological contribution to the field. In particular, the authors claim to use the spatial and temporal information on the samples in their inference. However, while they used the Chinese Restaurant Process to the include the temporal information in the analysis, it is not clear where and how the spatial information is used in their model. Also, comparison with the current approaches is very limited.

=== Major:
- How can your approach be used on multiple samples that vary only spatially? Do you assume that all samples are "sequential" in the sense the they are taken along a line? Discuss if and how can your approach be extended in 2 or 3 dimensions. Also, what if both time and space vary together?

- In the real dataset, how do you compare your results vs. MCMC? Did you use any gold-standard to determine which approach was actually better on the real dataset?

- There are several other approaches that have been developed to do a similar task including: Clomial, cloneHD, PhyloWGS, PyClone, Cloe, phyC, Canopy, TargetClone, ddClone, PASTRI, GLClone, TRaIT, WSCUnmix, and B-SCITE. A fair comparison with some of these alternative approaches is desirable.

=== Minor:

- The abstract is somewhat too technical. Please consider rewriting it for a broader range of audience especially biologists who are not necessarily interested much in the technical and methodological details. For example, explain why and how your sequential algorithm could be better.

- Some sentences are too long, e.g., lines 30-34, 40-44, etc.

-"These approaches have several limitations that prevent their wider usage in examining and quantifying the level of heterogeneity in a given sample." Briefly, mention these limitations.

- Typo: "112 SOME LATEX EXAMPLES"

- When referring the literature, please consider citing the original work. For example, a version of Eq. (2) has been mentioned and used in (Zare et al. 2014) before (Lee et al., 2016).

- Typo: "(Griffiths and Ghahramani, 2011; ?)"

- Typo: "then such a binary matrix is a draw from the distribution in (3) (?)"

- Line 154: Remove "due to limited space".

- Line 136: Remove "(discussed below)".

- Typo: 194 Reference to Figure ??.

- Typo in the supposed: "The CLL datasets in [3] for the three patients and the prostate cancer dataset are presented in Tables 21 - ??." Also, did you analyze and explain a prostate cancer dataset?!

- In the caption of Tables S1-3, explain how they are different from the tables in the main text.

Reviewer 3 ·

Basic reporting

In general, there are numerous grammar error and typo throughout the manuscript. For example, the sentence on page 9 line 230 has a grammar error.

Page 4 before equation (3), “Griffiths and Ghahramani, 2011; ?)” and Page 4 line 150 (?). There are many unexpected question marks all over the manuscript. Please correct them.

Page 3, line 112 and page 7, line 194 should be deleted.

The English language should be improved and partially rewritten.

Experimental design

Please add some instructions and examples to run the source code. I did not find any instruction on the github: https://github.com/moyanre/tumor_haplotypes

Validity of the findings

The authors claimed that the proposed SMC algorithm benefits from larger number of loci. But the haplotype error (ez) in table 2 increased from 0 to 0.008 of 1000 loci and 2000 loci respectively, which is exactly opposite to this conclusion. It would be better to make an explanation for this phenomenon.

In CLL003 data analysis part, the proposed SMC algorithm identifies 7 latent haplotypes, C2 and C6 are dominate haplotypes which is not consistent with Manual and Phylosub method, since Manual and Phylosub yield 5 haplotypes and only one of the haplotype dominate others(with ~0.8 proportion). How to interpret the difference? The inconsistency of estimated haplotype happened in CLL006 and CLL077 as well. The CLL data analysis part should be improved.

It will be great if the authors can compare the SMC method with some well known subclonal reconstruction methods such as PyClone.

Additional comments

This paper presents a novel sequential Monte Carlo (SMC) algorithm to solve the feature allocation model which characterizes tumor heterogeneity by latent haplotypes. Based on the simulated data, the proposed SMC algorithm provides more accurate compared to the state-of-the-art Markov chain Monte Carlo (MCMC) approach. An additional feature of the proposed algorithm is that newly observed VAFs data from next-generation sequencing (NGS) can be analyzed to improve existing estimates without re-analyzing the previous datasets, which improve the efficiency.
The effort to develop a more accurate and efficient algorithm to infer latent haplotypes is valuable.

---

## Round 0.2 · Minor Revisions

The only question/revision is with regard to PyClone. As noted by a reviewer, it was indicated that comparison was made to results from PyClone in the supplementary materials, but it is not obvious where this comparison is. Please clarify this.

Reviewer 1 ·

Basic reporting

The authors have addressed my concerns.

Experimental design

The authors have included competing approaches with similar formulations and subsequently strengthen their manuscript.

Validity of the findings

No concerns.

Additional comments

The edits and alterations have improved the manuscript and I have no further comment.

Reviewer 2 ·

Basic reporting

The presentation improved.

Experimental design

No comment.

Validity of the findings

No comment.

Additional comments

1- "we included, in the Supplementary Material, the results for PyClone"

I could not find PyClone results in the supp file that I attached. Am I looking at the right version?

2- The authors explained why they called their approach "sequential" in their reply. Did they include this explanation in the revised manuscript?

Annotated reviews are not available for download in order to protect the identity of reviewers who chose to remain anonymous.

Reviewer 3 ·

Basic reporting

no comment

Experimental design

no comment

Validity of the findings

no comment

Additional comments

The authors have addressed all of my comments and concerns in the revised version. I have no additional comments. Overall, the proposed method will benefit other researchers in the fields. I recommend to accept the paper.

---

## Author Rebuttal · Round 0.2

# Response to Editor's and Reviewers' Comments

Title: *Characterization of tumor heterogeneity by latent haplotypes: A sequential Monte Carlo approach*
Authors: Oyetunji E. Ogundijo and Xiaodong Wang
* * *
We thank the editor and the reviewers for the valuable comments made. We have now addressed these comments, and the paper has been improved accordingly. The changes in the revised manuscript are highlighted in red.
* * *
# Editor's comments:

Please consider and address the various reviewer comments. In particular, improvements to the writing/language should be incorporated. Comparison to other methods is also an important aspect.

**Response**: All the comments of the reviewers have been carefully addressed. The writing/language has been improved accordingly. All typos and errors have been corrected. As rightly pointed out by Reviewer 1, we have rigorously compared our method to one other method, MAP-based MAD proposed by Xu et al (JASA 2015), that also infers haplotypes and their content from tumor samples.

# Reviewer 1

## 1. Basic reporting

The English could be improved - some sentences are difficult to follow.
Some examples include:
In the abstract, the authors say "..and the parameters of our proposed state-state model." I assume they mean "state-space model."

Lines 52-54: Use of both "Although" and "however" makes this sentence confusing.
Line 82: authors should define "IBP" (Indian Buffet Process) in the main text.
Line 93: authors introduce matrices Z and W with no context or definition
Line 112: "SOME LATEX EXAMPLES"; I'm not sure why this is here.

Also, citation of Wersto et al. does not make sense (Line 41). The context involves studies that investigated tumor heterogeneity using NGS technology – Wersto et al. looked at tumor DNA content using flow cytometry. Additionally, the necessary depth of sequencing necessary to characterize somatic variation is quite high, making it rather dubious to claim such studies have been conducted for the "past few decades".

**Response**:

We have carefully reviewed and re-written the manuscript.

The abstract has been reviewed and all the typos have been corrected.

The sentence in Line 52 – 54 has been reviewed and re-written. This sentence is now in good shape.

We have defined IBP (Indian Buffet Process) the first time it appeared in the main text.

Matrices Z and W have been duly defined in the main text.

"SOME LATEX EXAMPLES" is a typo and it has been removed in the revised manuscript.

The citation Wersto et al has been removed and the paragraph has been re-written.

## 2. Experimental design

There are other similar approaches that were not addressed. The most obvious comparator would be the MAD-Bayes approach proposed by Xu et al. (JASA 2015), which similarly touts computational benefits over MCMC-based inference and applies IBP priors for feature allocation. It would seem important to compare against this approach. However, it is not mentioned in the manuscript and its unclear why the authors did not investigate.

**Response**:

In the revised manuscript, we have rigorously compared the MAP-based approach proposed by Xu et al (JASA 2015) to the two previously compared algorithms. All the three algorithms address inference of haplotype structure and their content in the tumor samples.

## 3. Validity of the findings

The methods appear well-outlined and the modes of evaluating error in the simulations seem reasonable.

**Response**:

We appreciate your important observation and comments.

# Reviewer 2

## 1. Basic reporting

Some typos must be fixed. Some sentences are too long.

**Response**:

Thanks for your important observation. All the typos have been fixed. The manuscript has been carefully re-written.

## 2. Experimental design

Will the code to reproduce your results be publicly available? How can other research teams apply your algorithm?

**Response**:

We implemented the proposed SMC algorithm in MATLAB. The MATLAB script is available for download. We provided a link for the code in the submission. The link is: https://github.com/moyanre/tumor_haplotypes

## 3. Validity of the findings

The findings on the synthetic data show some improvement over MCMC, but there are many alternative approaches to compare with. Also, no validation on the real data.

**Response**:

In the revised manuscript, we have provided comparison with another algorithm, a MAP-based algorithm that is designed to estimate haplotypes and their content in tumor samples.

## 4. Comments for the Authors

### A. Summary

The manuscript describes an approach to estimate the clonal structure of a tumor using multiple samples obtained from a single cancer that vary patient spatially or temporally. This work may have some methodological contribution to the field. In particular, the authors claim to use the spatial and temporal information on the samples in their inference. However, while they used the Chinese Restaurant Process to include the temporal information in the analysis, it is not clear where and how the spatial information is used in their model. Also, comparison with the current approaches is very limited.

**Response**:

We appreciate your important and valuable comment. The proposed SMC-based algorithm, by design, treats each sample, obtained either temporally or spatially as a 'sample'. The assumption is that each sample, either temporally or spatially obtained, consists of an unknown number of haplotypes. The proposed algorithm is sequential because for all samples, the count data at each locus is processed at each time-step of the algorithm.

## B. Major

How can your approach be used on multiple samples that vary only spatially? Do you assume that all samples are "sequential" in the sense that they are taken along a line? Discuss if and how can your approach be extended in 2 or 3 dimensions. Also, what if both time and space vary together?

In the real dataset, how do you compare your results vs. MCMC? Did you use any gold-standard to determine which approach was actually better on the real dataset?

There are several other approaches that have been developed to do a similar task including: Clomial, cloneHD, PhyloWGS, PyClone, Cloe, phyC, Canopy, TargetClone, ddClone, PASTRI, GLClone, TRaIT, WSCUnmix, and B-SCITE. A fair comparison with some of these alternative approaches is desirable.

### Response:

The proposed algorithm can process any number of samples, obtained either temporally or spatially.

We do not assume samples are sequential. The "sequential" in the proposed algorithm refers to the manner in which the count data is being processed. The proposed algorithm processes the count data of the loci *sequentially,* one locus at every time step. By design, how the samples are obtained will not have any effect on the performance or operation of the algorithm. So if both time and space vary together, they will not affect the performance of the algorithm. In summary, the sequential nature of the algorithm is related to how the data of each locus (for all the samples) is processed.

The SMC-based, MCMC-based and the MAP-based (included in the revised manuscript) algorithms produced similar results on the real tumor datasets. Real tumor datasets are often characterized with unknown haplotypes and this makes it challenging to determine which algorithm performs best. However, in our simulations, all the three methods for estimating latent haplotypes have the same underlying model.

Most methods estimate the clonal structure from tumor sample data. Only few methods have modeled tumor heterogeneity as containing latent haplotypes. In the revised manuscript, we have included the results of another algorithm, a MAP-based algorithm that also estimates latent haplotypes and their content from tumor samples. To compare the estimates of haplotypes with the mutational profiles of tumor subclones, we included, in the Supplementary Material, the results for PyClone and a Manual method on the real data.

## C. Minor

-The abstract is somewhat too technical. Please consider rewriting it for a broader range of audience especially biologists who are not necessarily interested much in the technical and methodological details. For example, explain why and how your sequential algorithm could be better.

- Some sentences are too long, e.g., lines 30-34, 40-44, etc.

-"These approaches have several limitations that prevent their wider usage in examining and quantifying the level of heterogeneity in a given sample." Briefly, mention these limitations.

- Typo: "112 SOME LATEX EXAMPLES"

- When referring the literature, please consider citing the original work. For example, a version of Eq. (2) has been mentioned and used in (Zare et al. 2014) before (Lee et al., 2016).
- Typo: "(Griffiths and Ghahramani, 2011; ?)"
- Typo: "then such a binary matrix is a draw from the distribution in (3) (?)"
- Line 154: Remove "due to limited space".
- Line 136: Remove "(discussed below)".
- Typo: 194 Reference to Figure ??.
- Typo in the supposed: "The CLL datasets in [3] for the three patients and the prostate cancer dataset are presented in Tables 21 - ??." Also, did you analyze and explain a prostate cancer dataset?!
- - In the caption of Tables S1-3, explain how they are different from the tables in the main text.

## Response:

1. The abstract has been re-written. Much of the technical details have been removed. We believe the current form of the abstract is suitable for a broad range of audience.
2. The long sentences in the manuscript have been re-written.
3. The limitations of the alternative methods for probing tumor heterogeneity have been listed in the revised manuscript.
4. Zare et al. 2014 has been duly cited as a method that used a variant of the feature allocation model before Lee et al. 2016.
5. All the typos related to missing citations have been addressed in the revised manuscript.
6. "due to limited space" in line 154 has been removed in the revised manuscript.
7. "(discussed below)" in line 136 has been removed in the revised manuscript.
8. The typo in line 194 has been fixed in the revised manuscript.
9. The statement about prostate cancer was a typo and this has been fixed in the revised manuscript.
10. The captions in the Tables in the Supplementary Material have been re-written in the revised manuscript.

# Reviewer 3

## 1. Basic reporting

In general, there are numerous grammar error and typo throughout the manuscript. For example, the sentence on page 9 line 230 has a grammar error.

Page 4 before equation (3), "Griffiths and Ghahramani, 2011; ?)" and Page 4 line 150 (?).
There are many unexpected question marks all over the manuscript. Please correct them.

Page 3, line 112 and page 7, line 194 should be deleted.

The English language should be improved and partially rewritten.

**Response**:

All the grammatical errors and typos have been fixed in the revised manuscript. Line 112 on page 3 and line 194 on page 7 have been deleted in the revised manuscript. All the missing citations have been provided in the revised manuscript. The manuscript has been re-written. Thanks for your valuable comment and observation.

## 2. Experimental design

Please add some instructions and examples to run the source code. I did not find any instruction on the github: https://github.com/moyanre/tumor_haplotypes

**Response**:

Instruction on how to run the MATLAB files has been included online.

## 3. Validity of finding

The authors claimed that the proposed SMC algorithm benefits from larger number of loci. But the haplotype error (ez) in table 2 increased from 0 to 0.008 of 1000 loci and 2000 loci respectively, which is exactly opposite to this conclusion. It would be better to make an explanation for this phenomenon.

In CLL003 data analysis part, the proposed SMC algorithm identifies 7 latent haplotypes, C2 and C6 are dominate haplotypes which is not consistent with Manual and Phylosub method, since Manual and Phylosub yield 5 haplotypes and only one of the haplotype dominate others(with ~0.8 proportion). How to interpret the difference? The inconsistency of estimated haplotype happened in CLL006 and CLL077 as well. The CLL data analysis part should be improved.

It will be great if the authors can compare the SMC method with some well known subclonal reconstruction methods such as PyClone.

**Response**:

Thanks for your valuable comment and observation. We stated that the estimate of the **proportions (matrix W)** will improve when the number of loci increases. This is consistent with the proportion error which is 0.0060 for 1000 loci and 0.0048 for 2000 loci. The differences observed in the estimates of haplotypes by the proposed SMC algorithm and the estimates of mutational profiles of tumor subclones by the Manual and PhyloSub are due to the modeling approach in both cases. The proposed SMC-based, the MCMC-based and the MAP-based (included in the revised manuscript) estimate latent haplotypes and their content in tumor samples (we define the haplotype as SNVs on the same homologous genome). On the other hand, both the manual approach and PhyloSub estimate the mutational profiles of subclones. The slight difference in the modeling assumptions in both cases is responsible for the observed differences in the results.

## 4. Comments for the Author

This paper presents a novel sequential Monte Carlo (SMC) algorithm to solve the feature allocation model which characterizes tumor heterogeneity by latent haplotypes. Based on the simulated data, the proposed SMC algorithm provides more accurate compared to the state-of-the-art Markov chain Monte Carlo (MCMC) approach. An additional feature of the proposed algorithm is that newly observed VAFs data from next-generation sequencing (NGS) can be analyzed to improve existing estimates without re-analyzing the previous datasets, which improve the efficiency. The effort to develop a more accurate and efficient algorithm to infer latent haplotypes is valuable.

**Response**:

Thanks for your valuable comments. We appreciate.

---

## Round 0.3 · accepted · Accept

Thank you for addressing the remaining concerns and congratulations again.